# Medical students' and educators' opinions of teleconsultation in practice and undergraduate education: A UK-based mixed-methods study

**Lisa-Christin Wetzlmair-Kephart**[1], **Andrew O'Malley**[2], **Veronica O'Carroll**[2]*

**1** College of Health Sciences, Glenville State University, Glenville, West Virginia, United States of America,
**2** School of Medicine, University of St Andrews, St Andrews, Scotland (United Kingdom)

* vo1@st-andrews.ac.uk

## Abstract

### Introduction

As information and communication technology continues to shape the healthcare landscape, future medical practitioners need to be equipped with skills and competencies that ensure safe, high-quality, and person-centred healthcare in a digitised healthcare system. This study investigated undergraduate medical students' and medical educators' opinions of teleconsultation practice in general and their opinions of teleconsultation education.

### Methods

This study used a cross-sectional, mixed-methods approach, utilising the additional coverage design to sequence and integrate qualitative and quantitative data. An online questionnaire was sent out to all medical schools in the UK, inviting undergraduate medical students and medical educators to participate. Questionnaire participants were given the opportunity to take part in a qualitative semi-structured interview. Descriptive and correlation analyses and a thematic analysis were conducted.

### Results

A total of 248 participants completed the questionnaire and 23 interviews were conducted. Saving time and the reduced risks of transmitting infectious diseases were identified as common advantages of using teleconsultation. However, concerns about confidentiality and accessibility to services were expressed by students and educators. Eight themes were identified from the thematic analysis. The themes relevant to teleconsultation practice were (1) The benefit of teleconsultations, (2) A second-best option, (3) Patient choice, (4) Teleconsultations differ from in-person interactions, and (5) Impact on the healthcare system. The themes relevant to teleconsultation education were (6) Considerations and reflections on required skills, (7) Learning and teaching content, and (8) The future of teleconsultation education.

**Data availability statement:** The primary data supporting the findings of this study are available at https://doi.org/10.17630/84eb74f3-e316-4618-a112-19b2f24377ac. These files include audio and text data. Additionally, data were imported into analytical software (SPSS and NVivo) for further processing. Public sharing of the thesis data is restricted in accordance with University regulations. Researchers interested in accessing the data may contact research-data@st-andrews.ac.uk. Data sharing will be contingent on the requester agreeing to handle the data appropriately and in compliance with all applicable local requirements.

**Funding:** The author(s) received no specific funding for this work.

**Competing interests:** The authors have declared that no competing interests exist.

## Discussion

The results of this study have implications for both medical practice and education. Patient confidentiality, safety, respecting patients' preferences, and accessibility are important considerations for implementing teleconsultations in practice. Education should focus on assessing the appropriateness of teleconsultations, offering accessible and equal care, and developing skills for effective communication and clinical reasoning. High-quality teleconsultation education can influence teleconsultation practice.

## Introduction

Telehealth uses technology synchronously or asynchronously to deliver, manage, coordinate, and evaluate health and well-being services remotely [1–4]. The utilised technologies cover a range of methods such as real-time remote encounters with healthcare professionals over email, telephone, or video-call, or the access to educational material or electronically stored healthcare information (for example in electronic health records) [5]. The integration of telehealth in healthcare practice and education can revolutionise the way future and present healthcare professionals learn, practice, and ultimately deliver patient care. The use of telehealth services will continue to shape the global healthcare landscape [6,7] as healthcare systems worldwide face challenges such as workforce shortage and the increased complexity of long-term conditions and multimorbidity. The COVID-19 pandemic put additional stress on healthcare systems [8] and prompted a surge in telehealth services offered to patients and the publication of telehealth policies and guidelines between 2020 and 2022 [9–12]. Telehealth could mitigate against, for example, limited human resources, equipment, and health facilities by accelerating access to information for healthcare professionals and increasing accessibility to services for various patient and population groups [13,14].

One essential sub-service of telehealth is teleconsultation, which can increase patients' accessibility to healthcare and reportedly allows the extension of healthcare services to remote and rural areas [1,15,16]. Teleconsultations are defined as direct, synchronous, or asynchronous consultations between healthcare professionals and patients using communication technologies such as phone or video calls, or emails [17]. Evidence-based evaluations have shown similar patient outcomes for in-person and teleconsultations in primary care and mental health services [18]. The use of teleconsultations has been shown to improve patient management, decrease referral rates and the number of medical procedures, and shorten diagnostic times [17]. Although patients have reported benefits of and satisfaction with teleconsultations [19–21], concerns have also been highlighted from the practitioners' perspectives such as increased workload based on an increase in demand for teleconsultations [20,22]. Additionally, in some instances, dissatisfaction with in-person consultations following an initial remote consultation has been reported from both patient and practitioner perspectives. Evaluations of teleconsultation platforms in the UK showed that healthcare professionals prefer to use teleconsultations only "for routine follow-up of chronic, stable conditions" over acute conditions [20,23]. Nevertheless, teleconsultations are often utilised in acute care settings such as out-of-hours services or clinical triaging for emergencies [24].

During the pandemic, universities utilised teleconsultation platforms to ensure that medical education continued during governmental lockdowns [25–27]. This was implemented out of necessity to ensure that students continued to have patient contact. The volume of teleconsultation experience is likely to have been a stark contrast to the limited teleconsultation education featured in health and social care programmes prior to the pandemic, as reported in

a systematic review by Wetzlmair et al. [28]. This review highlighted that good quality teleconsultation education is vital for encouraging effective teleconsultation practice. Previous studies have reported that teleconsultation education at the pre-registration or pre-qualification level ultimately ensures safe and high-quality remote healthcare for patients [22,29], as education initiatives can ensure that medical education meets the requirements and needs of the health and social care sector [30,31]. However, only tailored education increases the likelihood of successful implementation of teleconsultation services and benefits practitioners' knowledge, skills, and attitudes [32–34].

Thomas et al. [35] identified education and training of both future and current workforce as key to ensuring sustainable and safe teleconsultations. A skilled and competent healthcare workforce should not only use technology competently and effectively, but should also be able to assess potential implications of the used technology on patients' safety, access to health and social care, and ethical and legal consequences [36–39]. As the COVID-19 pandemic accelerated the use of teleconsultations across the health and social care sector [40,41] and the use of teleconsultation continues to expand, the education of present and future practitioners will become crucial. However, with limited educational frameworks and standardised teleconsultation integration into healthcare curricula [42], healthcare professionals may not be provided with the opportunities to build the competencies and confidence they need, to be able to work in a digitised healthcare system [30,35,43]. As teleconsultations are only sporadically integrated into most medical curricula [28], medical education and research need to investigate factors that inform the development and ultimately implementation of teleconsultation education.

This paper focuses on the findings related to two objectives of a larger study which aimed to investigate the factors that inform the development of teleconsultation education in undergraduate medical education in the UK. The two objectives focused on in this paper are:

a) to determine the opinions of undergraduate medical students and medical educators on teleconsultation applied in healthcare practice,

b) to determine the opinions of undergraduate medical students and medical educators on teleconsultation utilised in educational settings.

## Methods

This study used a cross-sectional, convergent mixed-methods study design (Fig 1) where both quantitative and qualitative data collection methods occurred simultaneously [44–46] to investigate undergraduate medical students' and medical educators' opinions of teleconsultation practice and education in the UK. This mixed methods model enables the researcher to address different aspects of the research questions and compare, contrast, and correlate results to provide more comprehensive information for the interpretation of the results [47,48]. The results from this study were analysed separately and integrated during data interpretation.

Both research design and research methods followed a mixture of constructivist and post-positivist approaches, with more emphasis put on the constructivist paradigm. A-priori decisions on sequencing priorities and assumptions on how to integrate quantitative and qualitative methods were made and influenced by the research paradigm.

### Development and administration of the research instruments

The development of the questionnaire was informed by the Technology Acceptance Model (TAM) [51] and findings from a systematic literature review related to commonly reported teaching methods [28]. Items were assessed for their validity and reliability [52] by three

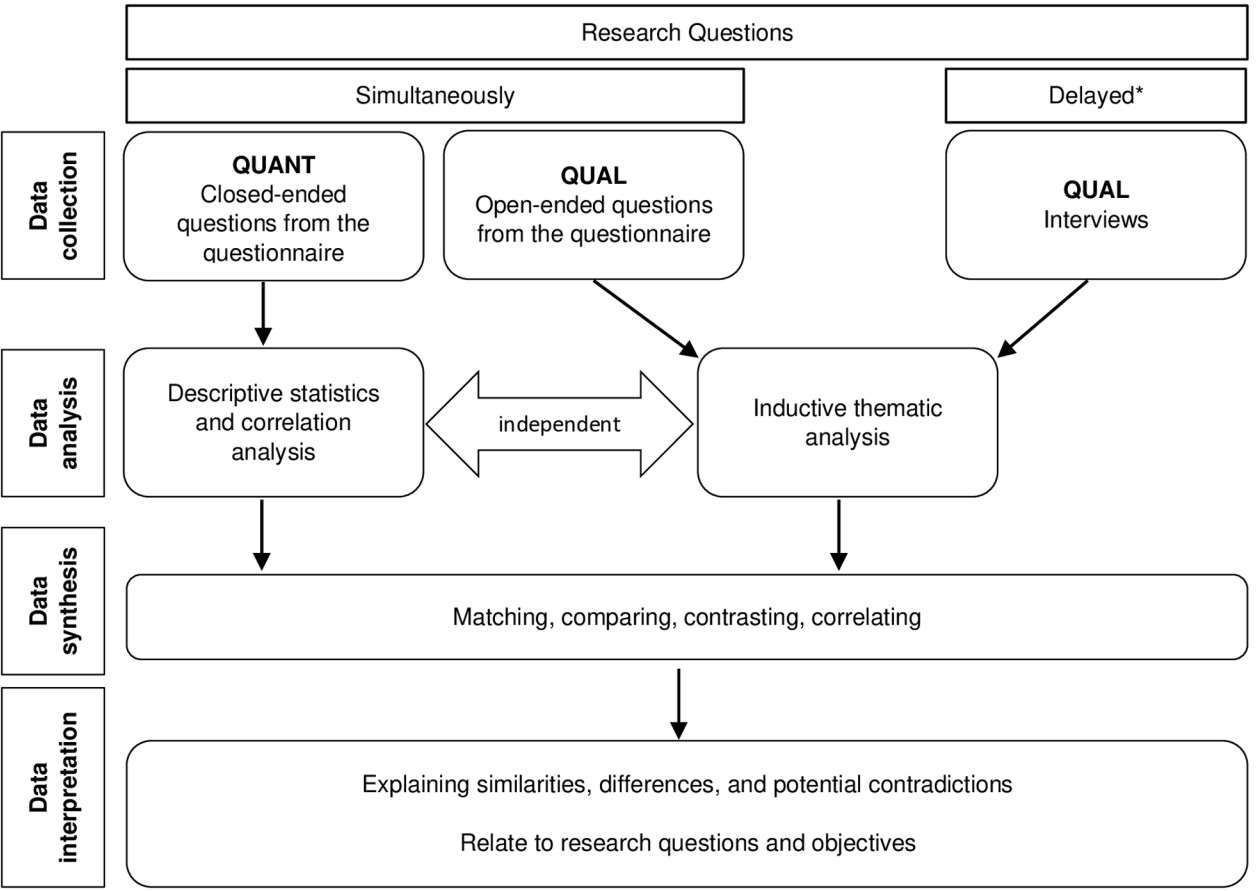

**Fig 1. Illustration of a convergent mixed-methods design with qualitative priority and quantitative input, based on** [ 47–50]. *Delay due to the optional sign-up after responding to the questionnaire.

independent researchers (VOC, AOM, L-CW-K) and were subject to a test-retest approach involving five medical students and five medical educators. The questionnaire was administered using the online software Qualtrics [53]. The final version of the questionnaire (S1 File) comprised a total of 29 questions. Display logic was introduced as not all questions were equally relevant to all participants. Most questions were closed, requiring participants to indicate a response from single or multiple-choice options and five-point Likert scale responses. These scales assessed the strength of agreement related to perceived usefulness and perceived ease of use of, attitudes towards, and difficulties in the use of teleconsultations. The final questions asked participants to rank the perceived advantages and disadvantages for patients and healthcare professionals using teleconsultations. The ranking items derived from the literature and are summarised in the S2 Table. The questionnaire ended with an open-ended question, enabling participants to provide further comments on the perceived advantages and disadvantages of teleconsultations in clinical practice and medical education.

A topic guide (S3 File) was used to guide the discussion in the semi-structured interviews. The topic guide was based on the research objectives, specifically the objective of gaining a more in-depth understanding of personal opinions of teleconsultation practice and medical education at UK medical schools. The questions were informed by qualitative research published in this field and related to previous learning and practical application of

teleconsultation [54] and the opinions on the implementation of teleconsultation in medical education [54,55]. The interviews were conducted remotely by one researcher (L-CW-K) utilising the software Microsoft Teams [56] between the 14th of June 2021 and the 16th of March 2022. A real-time analysis of the first ten interviews ensured that the interviewer (L-CW-K) adhered to the interview guide [57].

## Recruitment and sampling

Medical educators working in a clinical and/or academic setting in the UK and undergraduate medical students from all year groups studying at a UK medical school were eligible to participate in the study. The online questionnaire was distributed to medical students studying in university in the UK and to medical educators working within academic and/or healthcare settings in the UK between the 19th of May 2021 to the 13th of March 2022. After completing the questionnaire, participants were given the option to participate in a semi-structured interview. Written consent was obtained on the first page of the questionnaire. Interview participants consented to the interview via email prior to the interview.

The recruitment techniques for the quantitative aspect of the study were cluster, convenience, and snowball sampling [58]. For the initial sampling strategy, a cluster of all medical schools in the UK (to date 2020, n = 42) was created and a participant recruitment letter was sent to the Heads of Schools asking them to distribute the link to the questionnaire to their undergraduate medical students and medical educators affiliated with the respective school. In addition, separate emails were sent to the medical directors of Wales (n = 1) and Northern Ireland (n = 5) and to the Directors of Medical Education within Scottish health boards (n = 15). Convenience sampling referred to the sampling strategy utilising the social media platform X (formerly Twitter) to recruit contacts known to the research team. Those contacts ("gatekeeper", n = 7) were clinicians with involvement in undergraduate medical education across the UK. A snowball sampling technique was applied by encouraging gatekeepers and interview participants to distribute the questionnaire to their study and work colleagues. The qualitative sampling technique was based on self-selection. All study participants who participated in the online questionnaire were offered the option to participate in a semi-structured interview.

Data collection was initiated after the initial ethics approval received from the School of Medicine at the University of St Andrews (approval code MD15263). Two ethical amendments were approved to reflect changes to the initial recruitment strategy. Additional approval and permission to undertake the research was provided by the UK Medical Schools Council (MSC), and individual UK medical schools upon request.

## Data analysis

The quantitative data analysis was based on descriptive and correlation analyses. To align with the objectives of this paper, only a subsection of the questionnaire items was included in the quantitative analysis. These items included: demographical questions, ranking tasks regarding advantages and disadvantages perceived from a healthcare professional and patient perspective, and opinions of the inclusion of teleconsultation education within the UK medical curriculum. Participant characteristics were reported by measure of central tendency (mean, standard deviation) [59]. The Spearman-rank correlation method was applied to the categorical variables [60] and is subsequently reported with the coefficient $\rho$ (rho). The ranking items were analysed using frequencies and utilised a weighted average of the responses to account for the weight (i.e., importance) of the responses [61]. In a two-step approach, a rank score (RS) was calculated by multiplying the number of responses of each item with the "weight"

of the answer (e.g., 7 is the most important advantage, 1 is the least important advantage). This RS was then divided by the total number of responses. The quantitative data analysis was performed utilising the Statistical Package for Social Sciences (SPSS), version 26 [62], and Microsoft Excel [63].

The qualitative data included the verbatim transcribed interview data and responses to the open-ended items in the questionnaire. The data was analysed thematically, utilising an inductive, organic analysis, mainly guided by the school of reflexive thematic analysis [57]. After an initial coding phase, one author (L-CW-K) named, defined, and described codes in a codebook. This codebook was shared with two other researchers (VOC, AOM) who each coded six different transcripts independently to ensure transparency, reliability, and validity [57,64]. An inter-coder-reliability was calculated and a threshold of 80% agreement was defined. Agreement rates above 80% were not subject to further discussions. Rates below 80% were not accepted, hence differences in coding were investigated and discussed between the researchers until a consensus was found. Data analysis of the interview transcripts and the open-ended questionnaire item was conducted in NVivo [65].

## Results

### Quantitative analysis

A total of 240 questionnaire responses were considered for statistical analysis. Participant characteristics are summarised in Table 1.

Regarding the level of education, half the students reported that they had completed education at a high school or college level (55.6%). Forty-seven reported having a Bachelor's degree (30.7%) and 13 had a Master's degree (8.5%). Three participants had a PhD (2.0%). Half of the students were in their first (n = 36, 23.5%) or second year (n = 38, 24.8%) of medical education, 23 students (15.0%) and 31 students (20.3%) were in their third or fourth year, respectively. Year five and year six were attended by eleven (7.2%) and six (3.9%) students, respectively. Three (1.2%) students were intercalating when they completed the questionnaire. Amongst the medical educators, most participants had a Master's degree (n = 27, 31.0%). A further 17 (19.5%) and 10 (11.5%) had a PhD or MD, respectively. Medical educators working in both settings (n = 34) reported that they spent, on average, more hours in a healthcare setting ($\bar{x}$ = 23.89, sd = 15.42) than in an academic setting ($\bar{x}$ = 15.07, sd = 12.24).

Participants' opinions of the advantages and disadvantages of teleconsultations are summarised in Tables 2 and 3, respectively. The advantages reported by healthcare professionals were ranked by 155 participants. The two most important advantages included the reduction of transmitting contagious diseases ($\bar{x}$ = 5.37, RS = 833) and easier accessibility to remote populations ($\bar{x}$ = 5.23, RS = 810). The avoidance of unnecessary medical procedures was the only item ranked on average between second to last or last ($\bar{x}$ = 1.89, RS = 293). Disadvantages were ranked by 153 participants. The results were more bimodal than the perceived advantages: half of the items scored on average between first and second place: Experiencing technical problems and concerns about confidentiality were both rated the same and perceived as important disadvantages ($\bar{x}$ = 4.22, RS = 645). The most important disadvantage was the limitations in physical examination ($\bar{x}$ = 4.99, RS = 764). The perception that delivering difficult or upsetting news is harder was perceived as the last important ($\bar{x}$ = 2.32, RS = 355) disadvantage of teleconsultations. Advantages from a patient perspective were ranked by 146 participants. The perceived time-saving component was the most important advantage ($\bar{x}$ = 5.86, RS = 856). Similar to healthcare professionals, the avoidance of unnecessary medical procedures was the least important perceived advantage ($\bar{x}$ = 1.72, RS = 251). A total of 150 participants ranked the disadvantage from a patient perspective. Access to technology ($\bar{x}$ = 4.33, RS = 650) and potential

**Table 1.  Participants' characteristics stratified by their role in medical education (i.e., medical students and medical educators).**

| | | Medical students | | Medical educators | | Total | |
|---|---|---|---|---|---|---|---|
| | | n | % | n | % | n | % |
| **Gender** | Questionnaire (Interview) | 153 (13) | 63.75 (56.52) | 87 (10) | 36.25 (43.48) | 240 | 100 |
| | Female (Interview) | 118 (9) | 77.1 (69.23) | 47 (6) | 54.0 (60.00) | 165 (15) | 68.8 (65.22) |
| | Male (Interview) | 31 (4) | 20.3 (30.77) | 36 (4) | 41.4 (40.00) | 67 (8) | 27.9 (34.78) |
| | Non-binary | 1 | .7 | 2 | 2.3 | 3 | 1.3 |
| | Prefer not to say | 3 | 2.0 | 2 | 2.3 | 5 | 2.1 |
| **UK Country** | | 144 | 94.1 | 80 | 92.0 | 224 | 93.3 |
| | England (Interview) | 91 (6) | 59.5 (54.55) | 31 (5) | 35.6 (45.45) | 122 (12) | 50.8 (52.17) |
| | Northern Ireland | 0 | 0 | 2 | 2.3 | 2 | .8 |
| | Scotland (Interview) | 53 (7) | 34.6 (58.33) | 40 (5) | 46.0 (50.0) | 93 (11) | 38.8 (47.38) |
| | Wales | 0 | 0 | 5 | 5.7 | 5 | 2.1 |
| | England and Scotland | 0 | 0 | 2 | 2.3 | 2 | .8 |
| **Education** | | 152 | 99.3 | 78 | 89.7 | 230 | 95.8 |
| | High School | 46 | 30.1 | 0 | 0 | 46 | 19.2 |
| | College/ Further Education | 39 | 25.5 | 1 | 1.1 | 40 | 16.7 |
| | Bachelor's Degree | 47 | 30.7 | 16 | 18.4 | 63 | 26.3 |
| | Master's Degree | 13 | 8.5 | 27 | 31.0 | 40 | 16.7 |
| | PhD | 3 | 2.0 | 17 | 19.5 | 20 | 8.3 |
| | MD | – | – | 10 | 11.5 | 10 | 4.2 |
| | Other | 4 | 2.6 | 7 | 8.0 | 11 | 4.6 |
| **Year of Study** | | 5 | 3.27 | | | | |
| | Year 1 | 36 | 23.5 | | | | |
| | Year 2 | 38 | 24.8 | | | | |
| | Year 3 | 23 | 15.0 | | | | |
| | Year 4 | 31 | 20.3 | | | | |
| | Year 5 | 11 | 7.2 | | | | |
| | Year 6 | 6 | 3.9 | | | | |
| | Intercalating | 3 | 1.2 | | | | |

risks for miscommunication due to the limited use of non-verbal cues ($\bar{x}$ = 4.25, RS = 637) were the most important disadvantages. Unlike the ranking for healthcare professionals, from a patient perspective, confidentiality concerns were perceived as the least important ($\bar{x}$ = 1.82, RS = 273) disadvantage of teleconsultations.

Medical students' opinions of the implementation of teleconsultation in the medical curriculum are significantly positively correlated with previous exposure in clinical placements ($\rho$ = .205; p = .038); however, previous exposure in academic learning environments is insignificantly negatively correlated ($\rho$ = -.113; p = .225) with positive attitudes towards integrating teleconsultation in undergraduate medical education. There was no significant correlation between medical educators offering teleconsultations in a healthcare setting ($\rho$ = -.030; p = .825) and teaching teleconsultation in an academic setting ($\rho$ = .257; p = .051) and their inclination to implement teleconsultation in the medical curriculum.

Table 2. Results of the ranking item regarding the perceived advantages of teleconsultations from the perspective of healthcare professionals and patients. Scale from 1 (least important) to 7 (most important).

| Healthcare professionals N = 155 | 7 | 6 | 5 | 4 | 3 | 2 | 1 | Rank Score | Weighted average |
|---|---|---|---|---|---|---|---|---|---|
| Time | 29 | 27 | 33 | 22 | 19 | 16 | 9 | 716 | 4.62 |
| Cost | 8 | 13 | 19 | 32 | 33 | 30 | 20 | 536 | 3.46 |
| Transmission | 55 | 35 | 22 | 18 | 12 | 7 | 6 | 833 | 5.37 |
| Long-term conditions | 11 | 23 | 24 | 33 | 37 | 19 | 8 | 624 | 4.03 |
| Access to remote population | 42 | 40 | 26 | 19 | 17 | 8 | 3 | 810 | 5.23 |
| Interprofessional Collaboration | 7 | 14 | 27 | 22 | 24 | 39 | 22 | 528 | 3.41 |
| Unnecessary medical procedures | 3 | 3 | 4 | 9 | 13 | 36 | 87 | 293 | 1.89 |
| Patients N = 146 | | | | | | | | | |
| Time | 69 | 34 | 17 | 13 | 8 | 3 | 2 | 856 | 5.86 |
| Cost | 6 | 48 | 32 | 25 | 13 | 12 | 10 | 663 | 4.54 |
| Transmission | 41 | 23 | 32 | 24 | 15 | 4 | 7 | 741 | 5.08 |
| Long-term conditions | 3 | 13 | 30 | 33 | 29 | 29 | 9 | 535 | 3.66 |
| Underserved areas | 20 | 13 | 23 | 29 | 41 | 14 | 6 | 606 | 4.15 |
| Consult with multiple healthcare professionals simultaneously | 5 | 11 | 11 | 16 | 29 | 55 | 19 | 436 | 2.99 |
| Unnecessary medical procedures | 2 | 4 | 1 | 6 | 11 | 29 | 93 | 251 | 1.72 |

Table 3. Results of the ranking item regarding the perceived disadvantages of teleconsultations from the perspective of healthcare professionals and patients. Scale from 1 (least important) to 6 (most important).

| Healthcare professionals N = 153 | 6 | 5 | 4 | 3 | 2 | 1 | Rank Score | Weighted average |
|---|---|---|---|---|---|---|---|---|
| Technical problems | 27 | 44 | 43 | 21 | 10 | 8 | 645 | 4.22 |
| Physical examination | 73 | 39 | 22 | 9 | 6 | 4 | 764 | 4.99 |
| Resources | 8 | 7 | 31 | 47 | 40 | 20 | 448 | 2.93 |
| Trained | 3 | 9 | 18 | 27 | 48 | 48 | 360 | 2.35 |
| Confidentiality | 27 | 44 | 43 | 21 | 10 | 8 | 645 | 4.22 |
| Breaking bad news | 5 | 11 | 16 | 26 | 33 | 62 | 355 | 2.32 |
| Patients N = 150 | | | | | | | | |
| Technical problems | 30 | 25 | 32 | 23 | 23 | 17 | 565 | 3.77 |
| Limited non-verbal cues | 41 | 28 | 31 | 31 | 15 | 4 | 637 | 4.25 |
| Access to technology | 38 | 41 | 29 | 23 | 13 | 6 | 650 | 4.33 |
| Knowledge and competencies | 16 | 35 | 23 | 26 | 42 | 8 | 533 | 3.55 |
| Confidentiality | 6 | 5 | 6 | 14 | 27 | 92 | 273 | 1.82 |
| Breaking bad news | 19 | 16 | 29 | 33 | 30 | 23 | 492 | 3.28 |

## Qualitative Analysis

A total of 23 interviews were conducted and analysed. 13 interview participants were medical students and ten were medical educators. Nine students (69.23%) and six educators (60.00%) identified as female. Participants were mainly geographically located in England (n = 12, 52.17%) and Scotland (n = 11, 47.38%). A total of eight themes were identified inductively from the thematic analysis. Five themes were related to opinions of teleconsultation in clinical practice and three themes were related to teleconsultation education (Table 4).

**Table 4. Research objectives and identified themes.**

| Study objectives | Inductively identified themes |
|---|---|
| Opinions on teleconsultation practice | (1) The benefit of teleconsultations<br>(2) Patient choice<br>(3) A second-best option<br>(4) Teleconsultations differ from in-person interactions<br>(5) Impact on the healthcare system |
| Opinions on teleconsultation education | (6) Considerations and reflections on required skills<br>(7) Learning and teaching content<br>(8) The future of teleconsultation education |

**Opinions of teleconsultation practice.** *Theme 1: The benefits of teleconsultations:* As illustrated in the quotes below, participants highlighted the personal and professional benefits of a hybrid approach to teleconsultations, comprising of a mix of in-person and remote:

> *I think probably the best way to manage it [telehealth] is to do a mixture of both that suits the parent and the child, but it should be seen as an adjunct, I think, rather than a sort of primary mode of assessment and intervention. – Interview #2 (Medical Educator, both settings)*

> *… it's tricky and you gotta have the right sort of patient as well. I mean, some people just don't really bother open-up on the phone (…) at the rural practice I think it's got less of a place than in inner city because you've got internet problems, phone signal isn't there, infrastructure issues … as well. – Interview #13(Student)*

Teleconsultations were perceived to enable practitioners to work remotely and provide flexibility. Seeing patients in their home environment, including caregivers and family members in the consultation, and observing the interactions between caregivers and patients might reveal essential information, which can contribute to a high-quality, patient-centred healthcare management plan:

> *… but it you know what I mean I'd rather have Mary at home, the nurse doing that for them … rather than drag Mary to my clinic. …. I did get a fair few patients with dementia as well, so I see them with their carer so that they are in their home environment, I'm not removing them away from it so it becomes more safe and secure for them … – Interview #3 (Medical Educator, healthcare setting)*

*Theme 2: Patient choice*: Participants discussed the need for and benefit of patient choice between face-to-face or remote consulting modalities to suit patients' needs: some patients habituated to remote consultations during the COVID-19 pandemic which allowed them to choose the consultation method that provided flexibility and increased their autonomy.

> *I mean the other thing we teach students as well is patient preference and patient choice, which is why I think from the pandemic we're not gonna go back to where we were. – Interview #2 (Medical Educator, both settings)*

> *… giving the patient that autonomy to choose and be more involved in their healthcare, I think, is one of the main priorities 'cause everyone is all about the patient-centred care – Interview #14 (Student)*

*Theme 3: A second-best option*: Patient safety and accessibility were identified as some of the main drivers for perceiving teleconsultation services as the second-best option. Cognitive and

physical impairment and potentially differing levels of digital literacy amongst patients were considered as factors that may play a part in limiting accessibility to teleconsultations:

> *Without the ability to perform physical examinations, care is less safe. Pts [patients] will be exposed to more harmful scans and tests to make up for the lack of physical examination –* Questionnaire response #10 (Medical Educator, both settings)

> *… probably the hardest thing is actually the patient side of the spectrum, really … if you're dealing with a lot of older people or people who are not digitally literate, it makes things obviously also a lot harder, or if there's like learning difficulties at the other side or like aids like hearing aids you can't really account for that …* – Interview #8 (Student)

In addition, participants felt that some consultations with patients such as breaking bad news, discussing sensitive information, or where a patient needed to be examined were suitable for teleconsultations. Deciding before a consultation whether a teleconsultation would be an appropriate method to use over an in-person consultation was deemed important:

> *I think there are certain things that you don't want to discuss over the phone. So, difficult news or very difficult conversations can be challenging over the phone and again, if someone is trying to bring up a difficult subject, particularly around mental health, you need to a lot of your assessment will be what they are and what they look like, and it feels quite impersonal having these discussions on these topics without sitting face-to-face.* – Interview #20, (Student)

> *And I think being clear what you can and can't do on a videoconsultation. We can do reasonably well with a history... examination of the patient I think is quite limited. And so that will affect the type of consultation … Some things will be suitable for a videoconsultation some things will not.* – Interview #1 (Medical Educator, healthcare setting)

> *The ones [patients] that I send in to be seen face-to-face are the ones that should have been face-to-face in the first place so one of the things about telemedicine is that you select the right patients for your remote consultations.* – Interview #4 (Medical Educator, healthcare setting)

*Theme 4: Teleconsultations differ from in-person interactions*: Participants perceived communication as harder during teleconsultations compared to in-person consultations. Whilst students mentioned difficulties in building rapport and interpreting non-verbal communication cues, educators with experience in academic and clinical learning environments highlighted the differences in information the practitioners can gather:

> *… it's a lot easier if you're sitting with someone in a room to make them comfortable and relaxed, whereas if you're over Teams I don't know there's still that kind of barrier, it's not connected really. …* – Interview #19, (Student)

> *… they [clinicians, students] need to have an awareness that they're not getting the information or the whole picture that they probably need … really a crucial piece of information might be missing.* – Interview #2, (Medical Educator, both settings)

> *Doctor-patient relationship is strengthened by face-to-face meetings.* – Questionnaire response #35 (Medical Educator, both settings)

*Theme 5: Impact on the healthcare system*: Participants mainly perceived that teleconsultations did not save time in clinical practice and suggested that it might even introduce more complex working environments where hybrid appointments might duplicate work or increase the overall workload for practitioners:

> *A lot of patients end up needing to come in anyway for tests/examinations, so telecommunication prior to this seems like extra time used – Questionnaire response #71 (Student)*

> *So, it's very important not to conflate the ideas that more telephone consultation equals I can see more patients because that's totally wrong and at its worst telephone consultations done badly leads to more work because the patients are left unfulfilled. – Interview #3 (Medical Educator, both settings)*

Students perceived that teleconsultations are most likely to be successful if the coordination between medical practitioners and other members of the interprofessional team is effective. Furthermore, to ensure the safety and effectiveness of teleconsultations, primary and secondary care models should be integrated. Students mostly referred to the role of nurses or receptionists in coordinating and triaging patients and the interprofessional collaboration between specialist units and the patients' GPs (general practitioners):

> I *mean, triage systems run pretty well when they're nurse lead, I think, and sort of other staff, I don't think putting another burden onto GP is necessarily the way to do it because you've got them out there so that resource of a kind of other staff that are just to be able to do it most of the time … – Interview #13 (Student).*

> *I feel like a collaboration with local GPs for this kind of service [routine follow-ups] would make the online consult service much more efficient. – Questionnaire response #233 (Student)*

**Opinions of teleconsultation education in undergraduate medical education.** *Theme 6: Considerations and reflections on required skills*: Interview participants had varying opinions of whether teleconsultations should be utilised in undergraduate medical education. Participants disagreed on whether medical students and practitioners require different skills and competencies for teleconsultations than for in-person consultations. Some participants stressed that teleconsultation skills and competencies can be acquired regardless of dedicated education:

> *… in a way, I'm not sure how much training is needed or whether it's more that kind of experiential learning of just doing it and seeing what it's like. – Interview #4 (Medical Educator, healthcare setting)*

> *… it [teleconsultation training] would be helpful, definitely but then I don't know I feel like there's so many other like priorities, that it's something that you end up just picking up you know … – Interview #5 (Student)*

> *If I can pick it [teleconsultation] up, I'm sure the younger generation as they come through (…) they might not even need specific training because they will just adapt quite easily. – Interview #3 (Medical Educator, healthcare setting)*

Other participants, however, saw the utilisation of teleconsultation education as a requirement for medical practitioners to navigate a digitised healthcare system:

> *… I think it's actually a different skill (…) I'm just using the same history-taking style … that I start using in person, but it doesn't always necessarily work quite as well I don't think. – Interview #13 (Student)*

*I think there probably is specific work to be done about developing the telecare telehealth stuff as a skill in its own right, I am certain that the clinicians who do best on a ward round and that kind of thing are not always the same ones who are best doing remote consultations. – Interview #7 (Medical Educator, both settings).*

*I think students (…) can find that quite disruptive because they're already struggling a bit sometimes to get into the flow, and they're really focusing and trying their best to build a rapport, and if their phone suddenly goes dead … I think it throws them a bit, just like it throws me a bit, but I've got a bit more used to it. – Interview #6 (Medical Educator, both settings)*

*Theme 7: Learning and teaching content*: Medical educators in favour of teleconsultation education highlighted specific learning and teaching content such as the opportunities and limitations of high-quality, safe, patient-centred telehealth. It was highlighted that these discussions should cover the limited possibilities for physical examination, potential technical difficulties, and confidentiality concerns.

*The preparation for teleconsultations is significant. Learning about the 'red flags'/triggers for organising a follow-up F2F [face-to-face] appointment is still ongoing and I think that I am not expert at that yet. – Questionnaire response #70 (Medical Educator, both settings)*

*So, for example, somebody tries to show you a rash on a screen that is a bit blurry or pixelated and we just couldn't see anything at all. So, I think it's important that people [clinicians, students] don't feel pushed into making a decision on the basis of inadequate information I think that- that can be quite dangerous. – Interview #1 (Medical Educator, healthcare setting)*

Students discussed that it would be essential to be taught how to ensure professionalism and patient confidentiality:

*I think more so like confidentiality issues and so … if you leave a voicemail like can you say who's calling and like from where that kind of thing, so, I think it would be more issues about confidentiality … – Interview #15 (Student)*

Furthermore, both students and educators mentioned that teleconsultation education should comprise appropriate methods for documenting remote encounters with patients and learning how to navigate the virtual consultation environment:

*Documentation, I think, making sure that people you know are taking appropriate clinical notes so you know you need to record this consultation in exactly the same way as you would normally... What happens if that [connection] breaks down, what we gonna do. We can't see things properly, so we have a backup telephone number … so for example in Near Me, the patient's telephone number comes up on the screen, so you do have a note of it but if you lose the connection, you lose the telephone number. – Interview #1 (Medical Educator, healthcare setting)*

*Asking if the patient can see us, can hear us alright … these are basic things that we may not necessarily ask the patient if they're in person, but you would need to ask that if you're talking to them virtually, of course. – Interview #7 (Student)*

*Theme 8: The future of teleconsultation education*: The general opinion was that teleconsultations would continue to be an integral part of healthcare. It was highlighted that learning

should take place in both academic and clinical learning environments. Online placements should be offered to students, allowing them to actively practice remote consultations as opposed to observing clinicians consulting with patients remotely:

*(…) they've [students] maybe done some simulations in their universities first and they thought through the fact that they weren't going to be able to avoid telecommunication, then I think they might come with more informed questions about how to develop their practice … so, I think it's interesting to think about how they could come better prepared. – Interview #2 (Medical Educator, healthcare setting)*

*If I had the ability to include within my clinic the time to allow a student to make the phone call initially to then be able to discuss the case with them and call the family back or to take over the consultation to make the plan. I think it would give the students a real-life rather than virtual learning experience in terms of remote consultation. – Interview #2 (Medical Educator, healthcare setting)*

Interview responses indicated that teleconsultation education in both academic and clinical learning environments faces challenges. Whilst the medical curriculum is content-rich and adding teleconsultation education is not necessarily perceived as a priority by students, clinical placements that focus solely on teleconsultations might not be as educational for students and take away from other clinical skills:

*It [teleconsultation education] would be helpful, definitely but then I don't know I feel like there's so many other like priorities, that it's something that you end up just picking up you know … I think they probably just don't have enough time to squeeze it in, and it's considered low priority. – Interview #5 (Student)*

*But if you're watching somebody else do teleconsultation … it's really hard to concentrate on … you're watching someone on the phone, so dull to watch, but doing it is completely different as like … you're actually doing it, so it's a lot more interesting. – Interview #13 (Student)*

*All virtual consultations limits face to face clinical examination skills learning for students. – Questionnaire Response #15 (Medical Educator, healthcare setting)*

## Discussion

This mixed-methods study investigated the opinions of teleconsultation in clinical practice and the opinions of utilising teleconsultation in undergraduate medical education within the UK. Medical students and medical educators had diverse opinions of the use of teleconsultation in practice. Whilst remote consultations might allow patients to be flexible and ensure more person-centred care, a large percentage of participants were critical of the advantages of teleconsultation in terms of offering safe and high-quality care, building rapport, ensuring equal and easy access to healthcare, saving time, and being appropriate to all patients' health and social care needs. Even though some participants questioned the extent of education needed to acquire skills and competencies for teleconsultations, most participants highlighted the need for some teleconsultation education within the academic and clinical setting.

### Patient confidentiality, safety, and accessibility

The findings of this mixed-methods study contribute to the body of evidence hinting at mixed opinions of telehealth in medical practice regarding patient confidentiality, patient safety, and

accessibility to services [66–69]. This paper specifically investigated the opinions of medical students and medical educators in both clinical and academic practice. The results indicated that teleconsultations increase concerns about patients' confidentiality, even though the benefits of seeing patients in their home environment and increasing access to care have been highlighted as an advantage in both quantitative and qualitative data.

Newly qualified doctors are required to communicate with patients "respecting confidentiality and maintaining professional standards of behaviour" [70]. Living arrangements can introduce additional barriers to accessing telehealth services as privacy concerns (e.g., no private space for a teleconsultation) might impose challenges, particularly for people of lower socioeconomic status [71]. The potential risk of being overheard might distract patients and decrease the opportunities to share essential, sensitive information [44,72]. The findings in this study showed that participants' opinions of implementing teleconsultation education into medical education are ambiguous. As concerns around confidentiality were a key factor shaping those opinions, the literature discusses telehealth competencies, which acknowledge the importance of incorporating methods of ensuring patient confidentiality in teleconsultations into medical curricula [37–39]. Ethical medical practice in ensuring the patients' confidentiality and privacy is applicable to both in-person and telehealth practices; however, specific considerations in remote consultations must be made [37,38]. Experiences from clinical practice recommended discussions around ensuring privacy on patient, healthcare professional, clinical, and policy levels [73] to increase professionals' confidence in offering and consequently implementing safe and accessible teleconsultations [34]. Recommendations such as encouraging professionals and patients to use headphones, seeking a private room whenever possible, avoiding recording the consultations [73], and the use of platforms that comply with national regulations [74] are probably most appropriate for the UK undergraduate medical education context.

Within this study, patient-specific factors such as competencies in the use of technology, access to technologies, and digital literacy were perceived as important factors to consider in both clinical practice and medical education. These findings are in line with reports across telehealth literature which advocate for assessing patient factors to improve patient safety and satisfaction when engaging in remote consultations [21,44,68,69,75,76]. Practitioners need to assess the patients' competencies and confidence levels in using technologies for remote consultations to successfully tailor healthcare to the patients' needs [77,78]. In addition, the effective, safe use of teleconsultations is also dependent digital literacy [44,75,76]. Digital literacy refers to the skills required to retrieve and apply healthcare information and knowledge utilising technology [79]. Patients should have these abilities and the resources to engage in remote consultations. Healthcare professionals need a profound understanding of digital literacy and its importance for equal access to healthcare [80] to adequately address their patients' needs.

Ultimately, the use of teleconsultations should neither jeopardise patient safety and quality of care nor increase inequalities in healthcare for vulnerable population groups [77,78], as teleconsultations will gain importance around global healthcare systems. Initiatives that foster the population's digital skills and ultimately their ability to understand, retrieve, and act on information given in a remote consultation are essential to ensure an ongoing safe healthcare delivery in a digitised healthcare setting [81]. On a policy level, awareness should be raised that access to adequate and affordable technology is an important social determinant of health, influencing most aspects defined by Dahlgren and Whitehead [82].

## Clinical appropriateness and workflow

Study results indicate that teleconsultations might save time for patients and reduce risks of transmitting infectious diseases. However, study participants remarked on the need for

assessing the appropriateness of teleconsultations over in-person interactions. Previous authors have highlighted the need for assessing "patient and practice readiness and impact" [37] and "patient safety and appropriateness use of telehealth" [38]. Undergraduate education should therefore prepare students to be able to make this assessment of whether a teleconsultation is clinically appropriate or not. This includes but is not limited to evaluating the benefits and risks of consulting with patients remotely and understanding clinical indicators for transferring remote interactions to in-person consultations [37]. Providing high-quality, safe, effective, and efficient care utilising remote consultations requires medical practitioners to correctly recognise when patients might be in urgent need of emergency medical treatment [83]. Undergraduate medical education could help students develop these skills by increasing their awareness of different decision-making and clinical reasoning processes required during teleconsultations.

Additionally, the clinical environment should be ready for teleconsultations. As both the interviews and questionnaire responses indicated, introducing teleconsultations might increase the complexity of existing clinical workflows. Farr et al. [20] evaluated the implementation of a remote consultation system and found that medical practitioners felt frustrated by the duplication of work and additional time requirements when patients need to be seen in person following a remote consultation. Reportedly, the seamless integration of hybrid working arrangements that entail switching between in-person and remote consultations in clinical practice often constitutes a barrier and reduces uptake rates of teleconsultations [84]. And even though participants in this study perceived the closeness to the patients' living environments to be benefitting a person-centred approach, limitations in building rapport and communicating effectively and safely using technology were stressed as caveats or disadvantages. If those concerns remain unmet in medical education, patients' safety can be negatively impacted [85]. Teleconsultation education should therefore ensure that students are skilled in taking a history and adapting their assessment skills where necessary and possible [86]. This could be achieved through simulated teleconsultations or more opportunities for students to practice teleconsultations in clinical settings [85,86].

## Implications for medical education

The results indicated that teleconsultation should be included in medical education, but the implementation requires careful consideration of learning content, context, and outcomes. Quantitative and qualitative data have shown different results: As students who were exposed to teleconsultations in clinical practice were more likely to favour the implementation of teleconsultation education into medical education, medical practitioners are on average questioning the need to formally implement teleconsultation education. In the interviews, students were more sceptical about including additional content into already dense curricula. Nevertheless, the literature supports the assumption that training the workforce is fundamental to ensure safe, effective teleconsultations and to sustainably integrate remote consultations in hybrid clinical workflows [34,35,44]. Considering the most likely hybrid future of healthcare delivery, the medical curricula must imbed teleconsultation skills in both academic and clinical medical education settings [87]. The findings from this study could serve as helpful guidance for medical educators to consider the implementation of teleconsultation.

The content of teleconsultation education is dependent on the learning context. The findings of this study relate to the UK context, in which medical education is usually an undergraduate program; however, exemptions may apply regionally. Furthermore, postgraduate courses or continuous professional development courses and diplomas that specialise in telehealth more broadly often aim to develop effective leadership and/or influence policymaking. A UK example is

the postgraduate programme "Leading Digital Transformation in Health and Care for Scotland" [88]. Nevertheless, in the context of this study, results partially support previous suggestions to differentiate between "universal fundamentals" and "telehealth specifics" [89]. Whilst the former constitutes the education on medical principles such as high-quality, person-centered, ethical care, the latter specifically advocates for an adaptation of these principles in remote consultation settings [30,36,40,74,89–91]. By adding practical learning experiences in both simulated and clinical learning environments [25,28,92], future practitioners will be able to adequately analyse the benefits and risks of utilising teleconsultation, learn different strategies for communicating effectively and safely in remote settings, and explore discipline-specific applications of teleconsultations [77].

Continuous educational research is conducted to further develop evidence-informed education for undergraduate medical students including teleconsultation and the more broader realm of telehealth [28,92,93]. Discussion at medical schools and clinical placements should be encouraged including the impact of teleconsultations on patient confidentiality and safety, the accessibility to healthcare, the assessment of the clinical appropriateness of teleconsultations, and the implications on clinical workflows. In any case, the use of teleconsultation should be evaluated thoroughly to not unintendedly increase the digital divide in vulnerable population groups and therefore increase health inequalities [77].

## Study limitations

The findings provide a cross-sectional picture of undergraduate medical students' and medical educators' opinions of teleconsultation practice and education within the UK. However, the response rate of the online questionnaire was low. This might be explained by limitations of the sampling strategy, survey fatigue, or the stress on educators and students involved in the healthcare and education system during the peak of the COVID-19 pandemic. Additionally, the participants were unevenly spread across the four UK countries which impeded a justification for sub-group analyses. The snowball sampling techniques and self-selection applied to the semi-structured interviews could have limited data saturation as important stakeholders and participants may have withheld their opinions or may not have been reached by the online questionnaire in the first place. Furthermore, the period for data collection extended over 13 months and as the pandemic unfolded, changes in opinions might have occurred. However, an analysis of changes over time was not the aim of this study. Due to the exploratory nature of this mixed-methods study, in-depth interferential statistics that allowed an explanatory comparison between and adjustment for participants' characteristics such as age and gender were not prioritised. Future studies, teleconsultation education evaluations, or quality improvement projects could consider these variables in their data analysis plan.

## Conclusion

With the ongoing development of a digitised health and social care system, there is an increasing demand for a skilled workforce. The medical educators and medical students in this mixed-methods study perceived teleconsultation ambiguously, highlighting the advantages but also the fundamental disadvantages of teleconsultations in practice This ambiguity may be reflective of when the study took place during a time when teleconsultations were a necessity during COVID-19 restrictions. Teleconsultation should not be a replacement for in-person health and social care where the latter is needed, nor should teleconsultation education replace the teaching of in-person consultation skills. However, medical education should ensure that the future workforce is adequately prepared for effective and safe teleconsultation practice and also ensure that students are equipped with the knowledge to make choices around the appropriate use of remote consultations over in-person consultations.

## Supporting information

**S1 File. Administered questionnaire.** This questionnaire was distributed to UK undergraduate medical students and medical educators.
(DOCX)

**S2 Table. Ranking items.** Table showing the references that informed the ranking items used in the questionnaire.
(DOCX)

**S3 File. Script for interviewer.** This topic guide was used to guide the discussion in the semi-structured interviews.
(DOCX)

## Author contributions

**Conceptualization:** Lisa-Christin Wetzlmair-Kephart.

**Data curation:** Lisa-Christin Wetzlmair-Kephart.

**Formal analysis:** Lisa-Christin Wetzlmair-Kephart.

**Investigation:** Lisa-Christin Wetzlmair-Kephart.

**Methodology:** Lisa-Christin Wetzlmair-Kephart.

**Project administration:** Lisa-Christin Wetzlmair-Kephart.

**Supervision:** Andrew O'Malley, Veronica O'Carroll.

**Validation:** Andrew O'Malley, Veronica O'Carroll.

**Writing – original draft:** Lisa-Christin Wetzlmair-Kephart.

**Writing – review & editing:** Andrew O'Malley, Veronica O'Carroll.

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
