## [Decision Letter · Decision Letter 0]

2 Dec 2024

PONE-D-24-11694Medical students’ and educators’ opinions of teleconsultation in practice and undergraduate education: a UK-based mixed-methods studyPLOS ONE

Dear Dr. O'Carroll,

Thank you for submitting your manuscript to PLOS ONE. After careful consideration, we feel that it has merit but does not fully meet PLOS ONE’s publication criteria as it currently stands. Therefore, we invite you to submit a revised version of the manuscript that addresses the points raised during the review process.

We look forward to receiving your revised manuscript.

Kind regards,

Christoph Strumann

Academic Editor

PLOS ONE

Journal Requirements:

Additional Editor Comments:

The manuscript has been evaluated by two reviewers. While one Reviewer suggest to accept the manuscript, the other reviewer suggest that the manuscript needs to be revised before it could be accepted for publication.

Reviewers' comments:

Reviewer's Responses to Questions

**Comments to the Author**

1. Is the manuscript technically sound, and do the data support the conclusions?

Reviewer #1: Yes

Reviewer #2: Yes

2. Has the statistical analysis been performed appropriately and rigorously? 

Reviewer #1: Yes

Reviewer #2: Yes

3. Have the authors made all data underlying the findings in their manuscript fully available?

Reviewer #1: Yes

Reviewer #2: Yes

4. Is the manuscript presented in an intelligible fashion and written in standard English?

Reviewer #1: Yes

Reviewer #2: Yes

5. Review Comments to the Author

Reviewer #1: Very interesting and actual scientific article, showing pro and cons of tele consultation. I approved after read and debate with other colleagues. In fact I teach teleheatlh at the university and I agree with the authors about the conclusion and discussion.

Reviewer #2: This study investigated undergraduate medical students’ and medical educators’ opinions on teleconsultation practice in general, as well as their perspectives on teleconsultation education. The topic is interesting and emerging. The amount of data collected is sufficient for the purpose, and the methodological description is adequate. However, some revisions are necessary to improve the readability of the manuscript:

- The section “Opinions of teleconsultation practice” should be shortened and presented more concisely. In its current form, it is redundant and less suitable for a scientific paper compared to a PhD thesis.

- The role of the PhD should be highlighted in the authors' contributions rather than in the introductory phases of the work.

- Describe the age of the participants.

- The authors could highlight whether any sex-related differences emerged.

- In the introduction and/or discussion, it would be helpful to add references to recent manuscripts (e.g., Health Serv Res. 2024 Aug 2;24(1):885. doi: 10.1186/s12913-024-11365-6).

- How were the interview questions selected?

6. PLOS authors have the option to publish the peer review history of their article (what does this mean? ). If published, this will include your full peer review and any attached files.

**Do you want your identity to be public for this peer review?** For information about this choice, including consent withdrawal, please see our Privacy Policy .

Reviewer #1: No

Reviewer #2: No

---

## [Author Response · Author response to Decision Letter 0]

17 Dec 2024

Thank you to the reviewers and editor for the feedback on our manuscript. We would like to highlight our responses as follows:

Feedback from Reviewer 2:

1. The section “Opinions of teleconsultation practice” should be shortened and presented more concisely. In its current form, it is redundant and less suitable for a scientific paper compared to a PhD thesis.

 This section has been reduced as advised.

 We shifted parts of theme 4 to theme 3 to avoid repetition and make the information more succinct.

 In addition, we deleted 5 quotes and moved one quote to theme 6 (under opinions of teleconsultation education)

2. The role of the PhD should be highlighted in the authors' contributions rather than in the introductory phases of the work.

 We rephrased the sentence in the introduction, highlighting that this paper is part of a larger study: “This paper focuses on the findings related to two objectives of a larger study which aimed to investigate the factors that inform the development of teleconsultation education in undergraduate medical education in the UK.”

 Additionally, we added a statement below the “Author contributions”: “This study was part of the first author’s PhD supervised by VOC and AOM at the School of Medicine, University of St Andrews (Scotland).”

3. Describe the age of the participants.

 We did not collect information on the participants’ ages (neither in the questionnaire nor the interviews). We acknowledge that this might pose a limitation, however, due to the explorative nature of this study, we believe that this information would not add additional information relevant to exploring students’ and educators’ opinions of teleconsultation in medical education. We added a statement in the subsection “Study limitations” – see next comment.

4. The authors could highlight whether any sex-related differences emerged.

 This was – similar to point 3 – not the focus of this study. We flagged this in the subsection “Study limitations”: “Due to the exploratory nature of this mixed-methods study, in-depth interferential statistics that allowed an explanatory comparison between and adjustment for participants’ characteristics such as age and gender were not prioritised. Future studies, teleconsultation education evaluations, or quality improvement projects could consider these variables in their data analysis plan.”

5. In the introduction and/or discussion, it would be helpful to add references to recent manuscripts (e.g., Health Serv Res. 2024 Aug 2;24(1):885. doi: 10.1186/s12913-024-11365-6).

 Thank you for bringing this study from Italy to our attention. We acknowledge the fast-developing area of teleconsultation/telehealth research and added this and other references:

 Recommended study = reference #34: Marsilio, M., Calcaterra, V., Infante, G., Pisarra, M., & Zuccotti, G. (2024). The digital readiness of future physicians: Nurturing the post-pandemic medical education. BMC Health Services Research, 24(1), 885. https://doi.org/10.1186/s12913-024-11365-6

 Additional review = reference #21: Alashek, W. A., & Ali, S. A. (2024). Satisfaction with telemedicine use during COVID-19 pandemic in the UK: A systematic review. Libyan Journal of Medicine, 19(1), 2301829. https://doi.org/10.1080/19932820.2024.2301829

 Additional study = reference #87: Davies, L., Lawford, B., Bennell, K. L., Russell, T., & Hinman, R. S. (2023). Telehealth education and training in entry-to-practice physiotherapy programs in Australian universities: A qualitative study with university educators. Musculoskeletal Care, 21(2), 491–501. https://doi.org/10.1002/msc.1723

6. How were the interview questions selected?

 Thank you for highlighting the lack of reporting here. We added the following statement to the subsection “Development and administration of the research instrument”: “The questions were informed by qualitative research published in this field and related to previous learning and practical application of teleconsultation (54) and the opinions on the implementation of teleconsultation in medical education (54, 55).”

---

## [Decision Letter · Decision Letter 1]

8 Jan 2025

Medical students’ and educators’ opinions of teleconsultation in practice and undergraduate education: a UK-based mixed-methods study

PONE-D-24-11694R1

Dear Dr. O'Carroll,

We’re pleased to inform you that your manuscript has been judged scientifically suitable for publication and will be formally accepted for publication once it meets all outstanding technical requirements.

Kind regards,

Christoph Strumann

Academic Editor

PLOS ONE

Additional Editor Comments (optional):

Your revised manuscript has only been reviewed by one reviewer (Reviewer 2), as Reviewer 1 has already accepted your manuscript in its previous form.

Reviewers' comments:

Reviewer's Responses to Questions

**Comments to the Author**

1. If the authors have adequately addressed your comments raised in a previous round of review and you feel that this manuscript is now acceptable for publication, you may indicate that here to bypass the “Comments to the Author” section, enter your conflict of interest statement in the “Confidential to Editor” section, and submit your "Accept" recommendation.

Reviewer #2: All comments have been addressed

2. Is the manuscript technically sound, and do the data support the conclusions?

Reviewer #2: Yes

3. Has the statistical analysis been performed appropriately and rigorously? 

Reviewer #2: Yes

4. Have the authors made all data underlying the findings in their manuscript fully available?

Reviewer #2: Yes

5. Is the manuscript presented in an intelligible fashion and written in standard English?

Reviewer #2: Yes

6. Review Comments to the Author

Reviewer #2: The authors have adequately revised the work, addressing all the issues raised. The limitations have been discussed. The manuscript is acceptable in its current form.

7. PLOS authors have the option to publish the peer review history of their article (what does this mean? ). If published, this will include your full peer review and any attached files.

**Do you want your identity to be public for this peer review?** For information about this choice, including consent withdrawal, please see our Privacy Policy .

Reviewer #2: No

---

## [Editor Report · Acceptance letter]

PONE-D-24-11694R1

PLOS ONE

Dear Dr. O'Carroll,

I'm pleased to inform you that your manuscript has been deemed suitable for publication in PLOS ONE. Congratulations! Your manuscript is now being handed over to our production team.

Kind regards,

on behalf of

Dr. Christoph Strumann

Academic Editor

PLOS ONE